# GAN Training Acceleration Using Fréchet Descriptor-Based Coreset

**Yanzhe Xu** [1] [iD] **, Teresa Wu** [1,*]**, Jennifer R. Charlton** [2] **and Kevin M. Bennett** [3]

1 School of Computing and Augmented Intelligence and ASU-Mayo Center for Innovative Imaging, Arizona State University, Tempe, AZ 85281, USA; yanzhe.xu@asu.edu
2 Department of Pediatrics, Division Nephrology, University of Virginia, Charlottesville, VA 22908, USA; jrc6n@hscmail.mcc.virginia.edu
3 Department of Radiology, Washington University, St. Louis, MO 63130, USA; kmbennett@wustl.edu
* Correspondence: teresa.wu@asu.edu

**Abstract:** Generative Adversarial Networks (GANs) are a class of deep learning models being applied to image processing. GANs have demonstrated state-of-the-art performance in applications such as image generation and image-to-image translation, just to name a few. However, with this success comes the realization that the training of GANs takes a long time and is often limited by available computing resources. In this research, we propose to construct a Coreset using Fréchet Descriptor Distances (FDD-Coreset) to accelerate the training of GAN for blob identification. We first propose a Fréchet Descriptor Distance (FDD) to measure the difference between each pair of blob images based on the statistics derived from blob distribution. The Coreset is then employed using our proposed FDD metric to select samples from the entire dataset for GAN training. A 3D-simulated dataset of blobs and a 3D MRI dataset of human kidneys are studied. Using computation time and eight performance metrics, the GAN trained on the FDD-Coreset is compared against the model trained on the entire dataset and an Inception and Euclidean Distance-based Coreset (IED-Coreset). We conclude that the FDD-Coreset not only significantly reduces the training time, but also achieves higher denoising performance and maintains approximate performance of blob identification compared with training on the entire dataset.

**Keywords:** coreset; Fréchet Distance; training acceleration; blob identification; deep learning; GAN



## 1. Introduction

Generative Adversarial Networks (GANs), proposed by Ian Goodfellow [1] in 2014, are a class of deep learning models that are applied to image processing and consist of two sub-networks: a generator and a discriminator. The generator is trained to synthesize artificial versions of the original images, and a discriminator is employed to distinguish the artificial images from the real images. During the GAN training, both the generator network and the discriminator network interact with each other iteratively, resulting in generated images resembling as close as to the real images. GANs have demonstrated state-of-the-art performance in many applications including medicine, such as medical image generation [2], medical image-to-image translation [3], etc. Most recently, CycleGAN [4] has attracted great attention to translate unpaired images from one domain (known as source domain, e.g., one image modality) to another (known as target domain, e.g., a different image modality) by simultaneously training two sets of generators and discriminators, one for each domain. CycleGAN has been applied to organ segmentation [3], tumor detection [5], medical image denoising [6], medical image synthesis [7], blob detection [8], etc. In general, there are two types of CycleGAN-based models: (1) imaging-level translation; (2) object-focused image translation. For imaging-level translation, the model transfers whole images from one domain to another domain on a pixel or voxel basis. For example, Oulbacha et al. [9] proposed a pseudo-3D CycleGAN to synthesize CT images from MRI

for surgical guidance. Yang et al. [10] proposed a switchable CycleGAN with adaptive instance normalization to generate synthesized images for hypopharyngeal cancer diagnosis. For object-focused image translation, the model considers the domain knowledge from the target objects as added constraints to regularize the image translation to alleviate the geometric distortion issue. For example, Zhang et al. [11] utilized segmentation to derive a shape consistency constraint and applied it on CycleGAN to solve the shape distortion in cross-modality synthesis. Ma et al. [12] introduced illumination regularization and structure loss function for medical image enhancement. Xu et al. leveraged the geometric information of the target objects to assist the synthetic image rendering [8].

While the performance of CyleGAN-based models in these applications are promising, a common critique is that the model training is slow and often limited by computing resources. For example, the original CycleGAN [4,13] took 220 h of training with NVIDIA Titan X GPU on 10,000 2D paintings images. The BlobGAN [8], trained on 1000 3D blob images, took ~26 h for 50 epochs. One may argue that the training time can be reduced using advanced GPUs. However, such computing resources may not be readily available. To reduce the training time, researchers began exploring training using a subset instead of the whole dataset. Using false positive rate to measure the performance of generated image quality, Nuha et al. [14] proposed the DCGAN model and showed the training on subsets can lead to a false positive rate comparable to the training using the entire dataset with less computing time. Unfortunately, it was noted that the performance may be unstable for some experiments, as DCGAN led to very low false positive rate [14]. To maintain the quality of GAN's generated image during subset selection, DeVries et al. [15] proposed a novel instance selection approach based on manifold density of dataset. They removed the low-density regions to improve subsets' samples quality. Yet, this assumes that the low-density regions are noisy data region which may not always be true for medical images. Additionally, the computational time of this approach trained on ImageNet was reduced from 14.8 days to only 3.7 days [14], which still is considered to be computationally expensive even with advanced GPU power, for example, eight NVIDIA V100 GPUs in this case.

Recently, Coreset has attracted considerable attention on its applications including GAN. For example, an Inception embedding-based Coreset approach was proposed in [16] to accelerate GAN training. While this approach can be applied to the BlobGAN [8], with an Inception classifier being employed first to derive Inception embedding, we argue that additional computational cost occurs, which is against the goal of this study. In addition, the Inception embedding may be problematic to calculate the distance metric due to the fact medical images are known to be noisy. Motivated by object-focused image translation approaches, we introduce Fréchet Descriptor Distance (FDD) derived from object-related statistics to accelerate GAN's training. To demonstrate this idea, we focus on one specific type of imaging problem: blob images. This type of imaging has been utilized in a number of medical applications such as nuclei in 2D microscopy images [17] and glomeruli in 3D kidney cationic ferritin-enhanced MRI [18,19]. These images have some common characteristics: the number of blobs is large, and the shape of these blobs roughly follows Gaussian distribution. In this research, we introduce a new blob-based FDD to measure the distances between the image pair based on object (blob) statistics. The Coreset is then constructed using the new FDD metric on the image pairs, then the Coreset selection step can be taken out of the GAN training loop. To validate the performance of an out of loop FDD-Coreset for GAN training, we conduct two experiments. The first experiment is to identify the blobs in a 3D simulated blob image dataset where the locations of blobs are known. We choose the naïve random sampling method and the Inception and Euclidean Distance-based Coreset (IED-Coreset) [16] for comparison. Other than computation time, eight performance metrics are used including Peak Signal-to-Noise ratio (*PSNR*), Detection Error Rate (*DER*), Precision, Recall, F-score, Dice, Intersection over Union (*IoU*), and Blobness (*PB*). In the second experiment, we implement the FDD-Coreset on 3D human kidney images obtained from MRI. Since there is no ground truth available,

we compare against the above two methods using stereology results. Both experiments support the conclusion that FDD-Coreset can significantly accelerate the GAN training with comparable performance.

The remainder of the paper is organized as follows: Section 2 provides a review on related works. Section 3 describes our proposed FDD-Coreset in detail. Section 4 demonstrates the comparative results on the 3D synthetic images and 3D human kidney images. Finally, the conclusions are presented in Section 5.

## 2. Related Works

The overall objective of this research is to develop computational efficiency strategies using Coreset for GAN model training, with the blob identification problem as the use cases. In this section, we first provide a review on blob identification followed by Coreset.

### 2.1. Blob Identification

While large objects can often be automatically or semi-automatically isolated, small objects (blobs) are difficult to identify (detect and segment). Blobs can range in size and location in images. Examples of blobs include cells or nuclei in images from optical microscopy [17], glomeruli in cationic ferritin-enhanced magnitude resonance images (CFE-MRI) of the kidney [18,19]. Major challenges to identify small blobs include low image resolution and high image noise. The small blobs are often massive and can overlap together. To overcome the challenges of blob identification, a number of blob detectors have been developed for blob identification such as nuclei detection and cell detection, among which scale-space-based blob detectors have attracted great attention. For example, Kong et al. [20] proposed the generalized Laplacian of Gaussian (gLoG), which accurately detected blobs of various scales, shapes and orientations from histologic and fluorescent microscopic images. Zhang et al. developed Hessian-based Laplacian of Gaussian (HLoG) [21] and Hessian-based Difference of Gaussian (HDoG) [22] to automatically detect glomeruli in CFE-MRI. However, these blob detectors are not robust to noise [23], leading to high false positive rates. Recently, deep learning-based blob detector, U-Net joint with Hessian-based Difference of Gaussian (UH-DoG) [24] has been proposed to take advantage of the complementary properties of U-Net [25] to alleviate the over-detection issue of HdoG. However, UH-DoG may overlook large variations in blob size. As an extension of UH-DoG, another small blob detector using Bi-Threshold Constrained Adaptive Scales, BTCAS [26], is able to address this issue by searching local optimal DoG scale, which is adapted to the range of blob sizes to better separate touching blobs, and thus, the under-segmentation issue of U-Net is addressed. While UH-DoG and BTCAS detect blobs to some extent, there are limitations such as domain difference between public datasets and target dataset, geometric difference between small blobs, etc. To address these limitations, a denoising convexity-consistent Generative Adversarial Network (BlobGAN) [8] has been proposed for improved small blob identification. Experiments have shown that this blob detector could achieve high denoising performance and selectively denoise the image without affecting the properties of blobs. However, the training of the BlobGAN is slow and computationally expensive. In this research, we propose a Coreset-based approach to accelerate its training.

### 2.2. Coreset

In Coreset, a subset of the data is to be optimally selected such that a model trained on the selected subset will perform as closely as possible to a model trained on the entire dataset. Huang et al. [27] constructed Coreset to accelerate the computation of K-means and K-median clustering algorithms. Sener et al. [28] formulated the active learning problem as a Coreset selection problem where only core samples are to be labeled to improve the performance of Convolutional Neural Network (CNN) models. Coreset has also been adopted in GAN training. For example, Sinha et al. [16] developed a Coreset sampling approach to speed up GAN training. A pre-trained Inception classifier was first implemented to extract the Inception embedding from the whole dataset based on which

pairwise Euclidean distances were calculated. The Coreset is then derived for training in each iteration through comparing the distance between data samples. While this approach accelerates the GAN training to some extent, it is noted that the Coreset needs to be generated within each iteration of the training, in other words, within the training loop. From iteration to iteration, it is highly likely the same images will be selected for the Coreset which is computationally inefficient, especially considering the long computational time required to generate the Inception embedding. Additionally, the authors of [16] focused on the GAN model, taking a batch of images as Coreset for training, while CycleGANs utilize the Instance Normalization [29] as its normalization method for each layer, and thus, the training of CycleGAN often needs one image instead of a batch of images per iteration. Recognizing the potentials of Coreset and issues from deriving Coreset within the training loop as in [16], we argue that the training of GAN, in this study, CycleGANs, can be greatly accelerated if the Coreset selection is taken out of the training loop. We propose to implement Coreset as a pre-training step instead of a within-training process, and in this case, the Euclidian distance calculated from Inception embedding (derived during the GAN training) for Coreset construction in [16] does not apply.

## 3. Methods

In the proposed FDD-Coreset approach, given the group of descriptors of blob distribution from the entire blob images dataset, a Fréchet Descriptor Distance (FDD) is first derived to measure the difference between each pair of blob images; a Coreset is then selected from the entire dataset based on FDD to train a GAN model. To demonstrate the concept of an FDD-Coreset, the GAN of interest in this research is BlobGAN [8], a model designed for small blob identification. The source code of BlobGAN is available at: https://github.com/joshlyman/BlobGAN (accessed on 23 July 2022).

### 3.1. BlobGAN and Blob Descriptors

BlobGAN (Figure 1) consists of three steps: (1) 3D synthetic blobs are rendered using a 3D elliptical Gaussian function. The 3D blobs with respect to the corresponding masks and true images comprise the training input; (2) a 3D GAN is trained to denoise the images; (3) denoised 3D blobs are identified from the denoised images. The goal of 3D blob synthesis in Step 1 is to mimic the blobs (e.g., glomeruli in [30]) in real noisy blob images and provides the blob mask to fix the location of blobs. The image denoising in Step 2 adopts the domain translation architecture from CycleGAN [4], which consists of two generators and two discriminators. Given the 3D clean synthetic blob images as source domain and the 3D real noisy blob images as target domain, BlobGAN is to train a generator to learn the mapping from the source domain to the target domain and train another generator to learn the mapping from the target domain to the source domain iteratively, until the translated 3D noisy fake blob images approximate the 3D real noisy blob images and the translated 3D denoised blob images approximate the 3D clean bob images. The two discriminators are maintained to distinguish real images and translated images obtained from generators. The total loss of BlobGAN consists of adversarial loss, cycle consistency loss and identity loss adopted from CycleGAN [4] and an additional convexity consistency loss proposed by BlobGAN. Once image denoising is completed, Hessian convexity mask and blob mask generated from denoised blob images are used to derive the final blobs under a joint constraint.

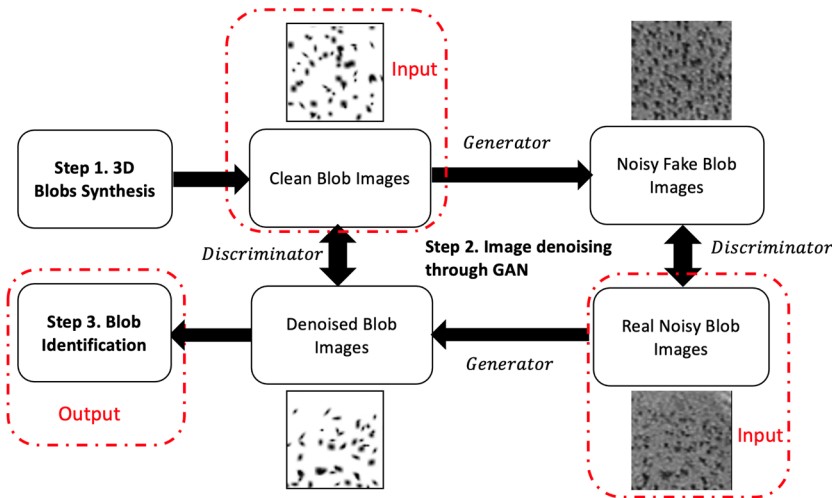

**Figure 1.** The overview of the BlobGAN model for blob identification.

Here, we apply the elliptical Gaussian function to identify the blob descriptors. A 3D elliptical Gaussian function is in the general form as follows:

$$F(x,y,z) = \mathcal{A} \cdot e^{-(a(x-x_0)^2 + b(y-y_0)^2 + c(z-z_0)^2 + d(x-x_0)(y-y_0) + e(y-y_0)(z-z_0) + f(x-x_0)(z-z_0))} \tag{1}$$

where $\mathcal{A}$ is a normalization factor, and $x_0$, $y_0$ and $z_0$ are the coordinates of the center of the Gaussian function $F(x,y,z)$. The coefficients $a$, $b$, $c$, $d$, $e$ and $f$ control the shape and orientation of $F(x,y,z)$ via $\theta$, $\varphi$, $\sigma_x$, $\sigma_y$ and $\sigma_z$ as given:

$$a = \frac{sin^2\theta cos^2\varphi}{\sigma_x^2} + \frac{sin^2\theta sin^2\varphi}{\sigma_y^2} + \frac{cos^2\theta}{\sigma_z^2}$$

$$b = \frac{cos^2\theta cos^2\varphi}{\sigma_x^2} + \frac{cos^2\theta sin^2\varphi}{\sigma_y^2} + \frac{sin^2\theta}{\sigma_z^2}$$

$$c = \frac{sin^2\varphi}{\sigma_x^2} + \frac{cos^2\varphi}{\sigma_y^2} d = \frac{sin2\theta cos^2\varphi}{\sigma_x^2} + \frac{sin2\theta sin^2\varphi}{\sigma_y^2} - \frac{sin2\theta}{\sigma_z^2} \tag{2}$$

$$e = -\frac{cos\theta sin2\varphi}{\sigma_x^2} + \frac{cos\theta sin2\varphi}{\sigma_y^2}$$

$$f = -\frac{sin\theta sin2\varphi}{\sigma_x^2} + \frac{sin\theta sin2\varphi}{\sigma_y^2}$$

Based on the 5 descriptors $\theta$, $\varphi$, $\sigma_x$, $\sigma_y$ and $\sigma_z$ from blob distribution, an FDD distance metric is proposed to measure the difference between each pair of images.

### 3.2. Fréchet Descriptor Distance

Fréchet Distance (FD) is used to measure the similarity between curves [31] and has been used in drug discovery [32], video applications [33], and others. In deep learning research, FD on Inception, termed Fréchet Inception Distance (FID) [34], is commonly used in GAN models. To compute FID, an Inception model $\mathcal{I}(\cdot)$ needs to be pre-trained, and the features from the penultimate layer of Inception model are extracted. Let $I_f : R^3 \to R$ be a 3D input image, and assuming these features from $I_f$ come from a multivariate Gaussian, the image features can be represented as:

$$\mathcal{I}\left(I_f; \Theta\right) \sim \mathcal{N}_f(\mu_f, \textstyle\sum_f), \tag{3}$$

where $\Theta$ is the parameter set of Inception model, and $\mu_f$ and $\sum_f$ are mean and covariance matrices of multivariate Gaussian $\mathcal{N}_f$. Let the image features on a 3D real image

$I_r : R^3 \to R$ be $\mathcal{I}(I_r; \Theta) \sim N_r(\mu_r, \sum_r)$ and the image features on a 3D synthesized image $I_s : R^3 \to R$ be $\mathcal{I}(I_s; \Theta) \sim N_s(\mu_s, \sum_s)$, then FID is defined as:

$$d_{FID}(N_r(\mathcal{I}(I_r; \Theta)), N_s(\mathcal{I}(I_s; \Theta))) = \|\mu_r - \mu_s\|_2^2 + tr(\sum_r + \sum_s - 2\sqrt{\sum_r \sum_s}), \quad (4)$$

where $tr$ is the trace of matrix. While FID is mainly used to evaluate the quality or effectiveness of GAN by measuring the similarity between real images and synthesized image, it has not been used as a distance metric in Coreset, mainly due to its computing costs. The computational complexity of FID is $\mathcal{O}(\mathcal{I}(I_r; \Theta) + \mathcal{I}(I_s; \Theta) + d_{FID})$. It is known that computing $\Theta$ and deep layers in Inception model is costly, with largest portion of computation dedicated for Inception model, that is, $\mathcal{O}(\mathcal{I}(I_r; \Theta) + \mathcal{I}(I_s; \Theta)) >> \mathcal{O}(d_{FID})$.

To accelerate GAN training, we propose Fréchet Descriptor Distance (FDD) as an alternative distance metric. This is motivated by the unique characteristics from the application of interest in this research, which is to say that the number of blobs is large and the shape of these blobs roughly follows Gaussian distributions. Instead of using FID on the whole images, FDD is to derive blob-related statistics which are used to measure the distances between the image pairs. In Section 3.1, we synthesize 3D blobs using 3D elliptical Gaussian function, and we know that each blob has 5 descriptors: $\theta$, $\varphi$, $\sigma_x$, $\sigma_y$ and $\sigma_z$. These blob descriptors are object-related statistics in 3D blob images. Assume each descriptor follows a Gaussian and a 3D blob image consist of $M$ blobs, the descriptors set $\Omega = \{\theta, \varphi, \sigma_x, \sigma_y, \sigma_z\}_{i \in M}$ follows a multivariate Gaussian. Let the descriptor set on a 3D blob image $I_b : R^3 \to R$ be $\Omega_b \sim N_b(\mu_b, \sum_b)$, then FDD for 2 pair-wisely compared 3D blob images $I_{b1}$ and $I_{b2}$ is defined as:

$$d_{FDD}(N_{b1}(\Omega_{b1}), N_{b2}(\Omega_{b2})) = \|\mu_{b1} - \mu_{b2}\|_2^2 + tr(\sum_{b1} + \sum_{b2} - 2\sqrt{\sum_{b1} \sum_{b2}}) \quad (5)$$

The computational complexity of FDD is $\mathcal{O}(d_{FDD})$. $\mathcal{O}(d_{FDD})$ is a function of $M$, the number of blobs and $\mathcal{O}(d_{FID})$ is a function of the image features from Inception model, for example, 64. We conclude $\mathcal{O}(d_{FDD}) \approx \mathcal{O}(d_{FID})$, $\mathcal{O}(\mathcal{I}(I_r; \Theta) + \mathcal{I}(I_s; \Theta)) >> \mathcal{O}(d_{FID}) \approx \mathcal{O}(d_{FDD})$, thus $\mathcal{O}(\mathcal{I}(I_r; \Theta) + \mathcal{I}(I_s; \Theta) + d_{FID}) >> \mathcal{O}(d_{FDD})$, and FDD is more computationally efficient for Coreset selection.

### 3.3. Dataset Sampling Based on Coreset

Models based on Coreset will provide approximate performance of using the whole dataset. Given an image dataset $D$, a Coreset $C$ of $D$ is a subset $C \subset D$ that approximates the "shape" of $D$. Let the cost function of model be $\mathcal{L}(\cdot)$, then the objective function to select Coreset can be represented as:

$$\min_{C:|C|=k} |\mathcal{L}(D) - \mathcal{L}(C)|, \quad (6)$$

where $k$ is desired size of $C$. Equation (6) indicates the performance under the selected Coreset $C$ with size $k$ to be as close as possible with the performance under the whole dataset $D$. We formulate it as a $k$-center problem (minimax facility location [35]) to choose $k$ core sample images such that the largest distance between a sample image and its nearest center is minimized. This is represented as:

$$\min_{C:|C|=k} \max_{I_i \in D} \min_{I_j \in C} d(I_i, I_j), \quad (7)$$

where $d(.,.)$ is a distance metric on $D$. This is an NP-Hard problem. However, it is possible to obtain an approximation solution efficiently using a greedy approach. Here, we use FDD as the distance metric $d(.,.)$ in Equation (7) and the details of an FDD-Coreset is shown in Algorithm 1.

---

**Algorithm 1:** Pseudocode for FDD-Coreset.

---

Input: target size $k$, 3D blob image datasets $D$ ($|D| > k$) with descriptors sets $\{\Omega_p\}_{p \in |D|}$

Output: Coreset $C$ ($|C| = k$)

1. Initialize Coreset $C = \{\}$

2. While $|C| < k$ :

3.       For 3D blob image $I_i$ from $D \backslash C$ :

4.       Extract the blob descriptors sets $\Omega_i$ of $I_i$

5.          For 3D blob image $I_j$ from $C$ :

6.          Extract the blob descriptors sets $\Omega_j$ of $I_j$

7.          Calculate FDD distance between $I_i$ and $I_j$: $d_{FDD}\left(N_i(\Omega_i), N_j(\Omega_j)\right)$

8.          Iteratively until find sample image $S = \underset{I_i \in D \backslash C}{\arg\max} \; \underset{I_j \in C}{\min} \; d_{FDD}\left(N_i(\Omega_i), N_j(\Omega_j)\right)$

9. Add sample image $S$ in Coreset: $C = C \cup \{S\}$

10. End While

11. Return Coreset $C$

---

## 4. Experiments and Results

### 4.1. Training Dataset

We evaluated our method using two experiments. The BlobGAN's training datasets in the source domain and target domain are different. In the source domain, we used the blob images synthesized by the 3D elliptical Gaussian function in Section 3.1 for both experiments. In the target domain, we used simulated noisy images in the first experiment and the real-world 3D human MR images of the kidney in the second experiment.

In the first experiment, we randomly synthesized 1000 3D blob images (64 × 64 × 32 voxels) to construct the source domain of the BlobGAN. The 3D training image (Figure 2a) blobs were scattered randomly in the image space. These blobs were generated using a random number generator and blobs' number ranged from 500 to 800. The parameters of 3D elliptical Gaussian function for each synthesized blob are as follows: $\theta$, $\varphi \in \left[0, 180°\right]$, $\sigma_x, \sigma_y, \sigma_z \in [0.5, 1.5]$. The blob mask (Figure 2b) was recorded for each blob image. For the target domain, we synthesized another 1000 3D blob images using the same 3D blob synthesis function. To simulate noisy 3D blob images (Figure 2c), we added random Gaussian noise to the synthesized images. The noise was generated by the Gaussian function, with $\mu_{noise} = 0$ and $\sigma^2_{noise}$ defined by:

$$\sigma^2_{\,noise} = \frac{\sigma^2_{\,image}}{10^{\frac{SNR}{10}}} \tag{8}$$

where the Signal-to-Noise Ratio ($SNR$) lies in the interval [0.01 dB, 1 dB]. The 1000 3D blob images and 1000 3D noisy blob images comprise the training dataset of baseline in the first experiment.

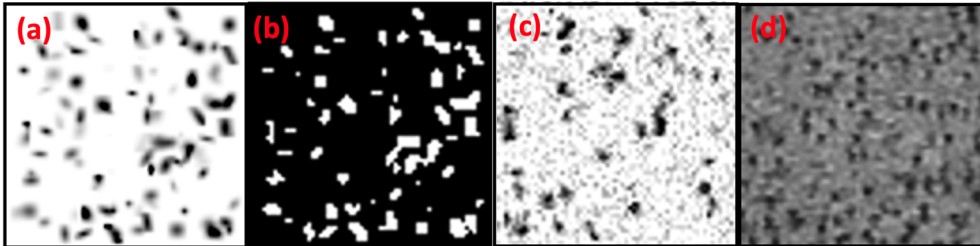

**Figure 2.** Illustration of training input images of BlobGAN (**a**) Synthesized 3D blobs image from source domain. (**b**) Blob mask of (**a**) from source domain. (**c**) Synthesized 3D noisy blobs image from target domain. (**d**) 3D MR image of the human kidney: patch from target domain.

In the second experiment, the source domain contains 1000 3D blob images (64 × 64 × 32 voxels). For target domain, we studied three 3D human kidneys MR images. Each

human kidney MR image has voxel dimensions of $896 \times 512 \times 512$. These three human kidneys were obtained after autopsy through a donor network (The International Institute for the Advancement of Medicine, Edison, NJ, USA) after receiving Institutional Review Board (IRB) approval and informed consent from Arizona State University [19]. They were imaged by as described in [19,36,37]. To validate our model, each time we trained the BlobGAN on two human kidneys and tested on the other one. We randomly sampled 1000 3D non-overlapping patch images ($64 \times 64 \times 32$ voxels, Figure 2d) from the two human kidneys in the training dataset. The sampling process was performed in the cortex region because the medulla region does not have any glomeruli. The medulla and cortex regions of human kidneys were annotated by a domain expert. The resulting 1000 3D blob images and 1000 3D human kidney patch images are treated as the training dataset of baseline in the second experiment.

### 4.2. Experiment I: Validation Experiments Using 3D Synthetic Image Data

To validate the performance of the FDD-Coreset on the BlobGAN, we synthesized an independent dataset of 1000 3D blob images ($64 \times 64 \times 32$ voxels) with Gaussian distributed noise as the test data. The version of these images without noise and their blob masks were recorded as ground truth to compare the performance with other approaches. The goal of this experiment is to show that the FDD-Coreset is capable of accelerating the training of the BlobGAN, while maintaining the BlobGAN's performance on blob identification. First, we selected $k$ Coreset from these 1000 3D blob images to train the BlobGAN. To train the BlobGAN, $\lambda_{cycle}$ and $\lambda_{convex}$ were set to 10, and $\lambda_{identity}$ was set to 0.5. The BlobGAN was trained from scratch using a learning rate of 0.0002 with the Adam optimizer (batch size = 1). The training typically took about 20 epochs to converge, so we did not set up the decay policy for the learning rate.

We compared the computation time of the IED-Coreset [16] with the proposed FDD-Coreset under varying sample size $k$ (see Table 1). As shown in Table 1, the FDD-Coreset is significantly faster (266–42,784 times) than the IED-Coreset for size $k$ from 1 to 10. We compared the total computation time from Coreset selection to training Coreset on the BlobGAN. We randomly sampled 1000 3D blob images to train the BlobGAN as baseline. We generated a Coreset using the IED-Coreset (size $k = 10$) and the FDD-Coreset (size $k = 10, 20, 30$), respectively, and trained these groups of Coreset on the BlobGAN by 20 epochs. Note that we did not evaluate the Coreset via the IED-Coreset with $k = 20$, 30. This is because with $k = 10$, the IED-Coreset took 42,784 times compared to the FDD-Coreset with $k = 10$. Once these Coresets were generated, we compared the total computational time from the Coreset generation to the BlobGAN training (see Table 2). It is apparent that the BlobGAN trained on the IED-Coreset with $k = 10$ takes the longest time. The reason is that the Coreset selection in the IED-Coreset takes almost 99% of the whole process. The BlobGAN trained on the FDD-Coreset with $k = 10, 20$ and 30 all performs significantly faster (31–87 times) than training on the entire dataset. This is because the FDD-Coreset reduces the entire 1000 3D images to $k$ core images which significantly reduces the BlobGAN training time. The BlobGAN trained on the FDD-Coreset with $k = 10, 20$ and 30 performs significantly faster (189–530 times) than training on the IED-Coreset with $k = 10$. The reason is that the IED-Coreset calculated the distance between images based on Inception embedding from images, and there exists duplicated computation in the batch selection. We conclude that the total computation time for training BlobGAN on the FDD-Coreset with $k = 10, 20$ and 30 is significantly less than the entire dataset and the IED-Coreset, even with $k = 10$.

**Table 1.** Computation time comparison of IED-Coreset and FDD-Coreset on 3D synthetic blob images (unit: seconds).

| $k$ Coreset Samples | IED-Coreset | FDD-Coreset |
|---|---|---|
| 1 | 2.66 | 0.01 |
| 2 | 3325.27 | 1.17 |
| 3 | 13,366.92 | 2.00 |
| 4 | 33,394.27 | 3.39 |
| 5 | 66,640.30 | 4.66 |
| 6 | 116,509.83 | 5.89 |
| 7 | 186,813.33 | 7.67 |
| 8 | 279,370.76 | 9.30 |
| 9 | 398,713.87 | 10.71 |
| 10 | 546,784.21 | 12.78 |

**Table 2.** Computation time comparison of BlobGAN trained on entire dataset (1000 random samples), IED-Coreset ($k = 10$) and proposed FDD-Coreset ($k = 10, 20, 30$) on 3D synthetic blob images (unit: seconds).

| $k$ Coreset Samples | Entire Dataset | IED-Corese | FDD-Coreset |
|---|---|---|---|
| 10 | | 547,825.60 | 1032.19 |
| 20 | 89,927.46 | – | 1995.25 |
| 30 | | – | 2896.44 |

We evaluated the model performance (blob identification) using eight metrics: Peak Signal-to-Noise Ratio (*PSNR*), Detection Error Rate (*DER*), Precision, Recall, F-score, Dice coefficient, Intersection over Union (*IoU*), and Blobness (*PB*), as follows:

1.  Peak Signal-to-Noise Ratio (*PSNR*) metric is to measure the performance of image denoising. Let the final 3D denoised blobs image be $x : R^3 \to R$ and the 3D blobs image without noises be $y : R^3 \to R$, then *PSNR* is defined as follows:

$$PSNR = 20 log_{10} \frac{MAX_x}{||x - y||_2}, \tag{9}$$

    where $MAX_x$ is the possible maximum voxel values of $x$.

2.  Detection Error Rate (*DER*) is to measure the difference ratio between the number of detected blobs and the ground truth. *DER* can be calculated by Equation (10).

$$DER = \frac{|N_{GT} - N_{Det}|}{N_{GT}} \tag{10}$$

    where $N_{GT}$ represents the # of ground truth blobs and $N_{Det}$ represents the # of detected blobs.

To avoid duplicate counting, the number (#) of true positives $TP$ was calculated by Equation (11)

$$TP = \min\left\{ \#\left\{ (i, j) : \min_{i=1}^{m} D_{ij} \leq d_\delta \right\}, \#\left\{ (i, j) : \min_{j=1}^{n} D_{ij} \leq d_\delta \right\} \right\}, \tag{11}$$

where $m$ is the number of true glomeruli, $n$ is the number of blob candidates, and $d_\delta$ is a thresholding parameter set to a positive value $(0, +\infty)$. If $d_\delta$ is small, fewer blob candidates are counted, since the distance between the blob candidate centroid and ground-truth should be small. If $d_\delta$ is too large, more blob candidates are counted. A candidate was considered as $TP$ if the centroid of its magnitude was in a detection pair $(i, j)$ for which the nearest ground truth center $j$ had not been paired, and the Euclidian distance $D_{ij}$ between ground truth center $j$ and blob candidate $i$ was less than or equal to $d_\delta$. Here, since local

intensity extremes could be anywhere within a small blob with an irregular shape, we set $d$ to the average diameter of the blobs: $d_\delta = 2 \times \sqrt{\frac{\sum_{(i,j,k)} f(i,j,k)}{\pi}}$. TP is used for the precision in Equation (12), Recall in Equation (13) and F-score in Equation (14).

3. Precision is to measure the fraction of retrieved blobs confirmed by the ground truth.

$$\text{Precision} = \frac{TP}{n} \tag{12}$$

4. Recall is to measure the fraction of ground-truth data retrieved.

$$\text{Recall} = \frac{TP}{m} \tag{13}$$

5. F-score is the overall performance of precision and recall.

$$\text{F-score} = 2 \times \frac{\text{Precision} \times \text{Recall}}{(\text{Precision} + \text{Recall})} \tag{14}$$

6. Dice coefficient (*Dice*) is to measure the similarity between the segmented blob mask and the ground truth.

$$Dice\ (BM, BG) = \frac{2|BM \cap BG|}{|BM| + |BG|}, \tag{15}$$

   where $BM$ is the binary mask for segmentation result and $BG$ is the binary mask for the ground truth.

7. Intersection over Union (*IoU*) is to measure the amount of overlap between the segmented blob mask and the ground truth.

$$IoU\ (BM, BG) = \frac{BM \cap BG}{BM \cup BG}, \tag{16}$$

   where $BM$ is the binary mask for segmentation result and $BG$ is the binary mask for the ground truth.

8. Blobness (*PB*) is to measure the likelihood of the objects with a blob shape. *PB* for each blob candidate $bi$ from blobs set $S_{blob}$ is calculated by Equation (17):

$$PB_{bi \in S_{blob}} = \frac{3 \times |\det(H(J - f))|^{\frac{2}{3}}}{pm(H(J - f))} \tag{17}$$

   where $f$ is the normalized 3D dark blobs image, $H$ represents the Hessian matrix and $pm$ represents the principal minors of Hessian matrix.

The performance comparison between the BlobGAN trained on the entire dataset (1000 random samples), IED-Coreset ($k = 10$) and FDD-Coreset ($k = 10, 20, 30$) is shown in Table 3. Compared to the BlobGAN trained on the entire dataset, the BlobGAN trained on the FDD-Coreset ($k = 20$) provides better performance on *PSNR*, and the BlobGAN trained on the FDD-Coreset ($k = 30$) provides better performance on *DER*, with comparable performance on Recall and F-score. The BlobGAN trained on the IED-Coreset ($k = 10$) and the FDD-Coreset ($k = 10, 20, 30$) gives, in both cases, a lower performance on Dice and *IoU* than the BlobGAN trained on the entire dataset, but the FDD-Coreset ($k = 30$) gives the closest performance. The BlobGAN trained on the FDD-Coreset ($k = 10$) has the closest *PB* value with the BlobGAN trained on the entire dataset and the ground truth. We conclude that the BlobGAN trained on the FDD-Coreset provides comparable performance to the model trained on the entire dataset.

**Table 3.** Performance comparison (avg ± std) of BlobGAN trained on entire dataset (1000 Random Samples), IED-Coreset ($k = 10$) and proposed FDD-Coreset ($k = 10, 20, 30$) on 3D synthetic images.

| Metrics | Entire Dataset | IED-Coreset ($k = 10$) | FDD-Coreset ($k = 10$) | FDD-Coreset ($k = 20$) | FDD-Coreset ($k = 30$) |
|---|---|---|---|---|---|
| *PSNR* | 12.539 ± 0.143 | 13.000 ± 0.753 | 12.084 ± 0.423 | 15.758 ± 0.373 | 11.489 ± 0.216 |
| *DER* | 0.091 ± 0.037 | 0.124 ± 0.131 | 0.171 ± 0.156 | 0.219 ± 0.061 | 0.052 ± 0.037 |
| Precision | 0.941 ± 0.013 | 0.759 ± 0.069 | 0.671 ± 0.056 | 0.923 ± 0.018 | 0.827 ± 0.024 |
| Recall | 0.855 ± 0.033 | 0.781 ± 0.052 | 0.771 ± 0.042 | 0.720 ± 0.049 | 0.845 ± 0.032 |
| F-score | 0.896 ± 0.018 | 0.766 ± 0.030 | 0.715 ± 0.028 | 0.807 ± 0.029 | 0.835 ± 0.014 |
| Dice | 0.825 ± 0.017 | 0.548 ± 0.046 | 0.662 ± 0.014 | 0.502 ± 0.039 | 0.671 ± 0.022 |
| *IoU* | 0.702 ± 0.024 | 0.379 ± 0.044 | 0.495 ± 0.015 | 0.336 ± 0.035 | 0.505 ± 0.025 |
| *PB* (Ground Truth: 0.519) | 0.538 ± 0.279 | 0.609 ± 0.302 | 0.577 ± 0.308 | 0.583 ± 0.290 | 0.610 ± 0.298 |

*4.3. Experiment II: Validation Experiments Using 3D Human Kidney MR Images*

In this experiment, we investigated the proposed FDD-Coreset approach on 3D MR images to measure the number ($N_{glom}$) and apparent volume ($aV_{glom}$) of glomeruli in healthy and diseased human donor kidneys that were not accepted for transplant. Since we have three 3D MR human kidney images, and each of them has voxel dimensions of $896 \times 512 \times 512$ and the input of the BlobGAN is a 3D patch with voxel dimensions of $64 \times 64 \times 32$, we divided each human kidney into 1792 3D patches ($64 \times 64 \times 32$) to validate the performance of the BlobGAN. The final identification mask of the whole kidney was reconstructed by stacking all 3D patches. To train the BlobGAN, $\lambda_{cycle}$ and $\lambda_{convex}$ were set to 10, $\lambda_{identity}$ was set to 0.5, the learning rate was 0.0002, and the Adam optimizer was used with a batch size set to 1.

We compared the total computation time from Coreset selection to training Coreset on the BlobGAN on 3D MR human kidney images. Similarly with Section 4.2, we have the original random sampled 1000 3D blob images to train the BlobGAN as the baseline, so we generated Coreset (size $k = 10$) through the IED-Coreset and the FDD-Coreset (size $k = 10$), respectively, and trained these groups of Coreset on the BlobGAN by 20 epochs. Once these Coreset are generated, we compare the total computational time from the Coreset generation to the BlobGAN training and summarize them in Table 4, from where we can see the results show that the BlobGAN trained on the IED-Coreset with $k = 10$ on three human kidneys takes the longest time. The BlobGAN trained on the FDD-Coreset with $k = 10$ on three kidneys all perform significantly faster (76 times) than training on the entire dataset. The BlobGAN trained on the FDD-Coreset with $k = 10$ on three kidneys performs, in all cases, significantly faster (460 times) than training on the IED-Coreset with $k = 10$. We can conclude that training the BlobGAN on the FDD-Coreset with $k = 10$ on 3D MR human kidney images is significantly computationally more efficient than the entire dataset and the IED-Coreset with $k = 10$, and this is consistent with the comparison of computation time in Section 4.2.

**Table 4.** Computation time comparison of BlobGAN trained on entire dataset (1000 random samples), IED-Coreset ($k = 10$) and proposed FDD-Coreset ($k = 10$) on 3D human kidney images (unit: seconds).

| Human Kidney | Entire Dataset | IED-Coreset ($k = 10$) | FDD-Coreset ($k = 10$) |
|---|---|---|---|
| CF 1 | 91,325.20 | 547,800.21 | 1195.58 |
| CF 2 | 91,382.40 | 547,801.81 | 1180.38 |
| CF 3 | 91,515.60 | 547,772.61 | 1199.58 |

$N_{glom}$ and mean $aV_{glom}$ are reported in Tables 5 and 6, where the BlobGAN trained on the entire dataset (1000 random samples), the BlobGAN trained on the IED-Coreset with $k = 10$ and the BlobGAN trained on the FDD-Coreset with $k = 10$ are compared to the data from unbiased dissector–fractionator stereology [18]. We used these stereology data from [19] as ground truth and calculated $aV_{glom}$ based on the method from [37]. The

differences between the results of these three training approaches and stereology data are also listed in Tables 5 and 6. As shown in Tables 5 and 6, $N_{glom}$ and mean $aV_{glom}$ derived from the BlobGAN trained on the FDD-Coreset with $k = 10$ provides a smaller difference ratio with stereology than the BlobGAN trained on the IED-Coreset with $k = 10$. While the BlobGAN trained on the entire dataset has the smallest difference ratio with stereology, it takes 96% more computational time than the FDD-Coreset with $k = 10$, as shown in Table 4. We could conclude that the FDD-Coreset not only significantly reduces the training time, but also maintains the approximate performance of glomerular segmentation compared with training on the entire dataset.

**Table 5.** Human kidney glomerular segmentation results ($N_{glom}$) using BlobGAN trained on entire dataset (1000 Random Samples), IED-Coreset ($k$ = 10) and proposed FDD-Coreset ($k$ = 10) comparing with stereology.

| HUMAN KIDNEY | $N_{glom}$ ($\times 10^6$) (STEREOLOGY) | $N_{glom}$ ($\times 10^6$) (ENTIRE DATASET) | Difference Ratio (%) | $N_{glom}$ ($\times 10^6$) (IED-CORESET ($k$ = 10)) | Difference Ratio (%) | $N_{glom}$ ($\times 10^6$) (FDD-CORESET ($k$ = 10)) | Difference Ratio (%) |
|---|---|---|---|---|---|---|---|
| CF 1 | 1.13 | 1.05 | **7.08** | 1.58 | **39.82** | 1.26 | **11.50** |
| CF 2 | 0.74 | 0.71 | **4.05** | 0.96 | **29.73** | 0.78 | **5.41** |
| CF 3 | 1.46 | 1.48 | **1.37** | 1.77 | **21.23** | 1.53 | **4.79** |

**Table 6.** Human kidney glomerular segmentation results (mean $aV_{glom}$) using BlobGAN trained on entire dataset (1000 Random Samples), IED-Coreset ($k$ = 10) and proposed FDD-Coreset ($k$ = 10) comparing with stereology.

| HUMAN KIDNEY | Mean $aV_{glom}$ ($\times 10^{-3}$mm$^3$) (STEREOLOGY) | Mean $aV_{glom}$ ($\times 10^{-3}$mm$^3$) (ENTIRE DATASET) | Difference Ratio (%) | Mean $aV_{glom}$ ($\times 10^{-3}$mm$^3$) (IED-CORESET ($k$ = 10)) | Difference Ratio (%) | Mean $aV_{glom}$ ($\times 10^{-3}$mm$^3$) (FDD-CORESET ($k$ = 10)) | Difference Ratio (%) |
|---|---|---|---|---|---|---|---|
| CF 1 | 5.01 | 5.19 | **3.59** | 4.48 | **10.58** | 5.20 | **3.79** |
| CF 2 | 4.68 | 4.80 | **2.56** | 3.61 | **22.86** | 5.02 | **7.26** |
| CF 3 | 2.82 | 2.81 | **0.35** | 2.25 | **20.21** | 2.93 | **3.90** |

For illustration, example results from CF1 are shown in Figure 3. As seen, the BlobGAN trained on the FDD-Coreset has more similar glomerular segmentation results to the BlobGAN trained on the entire dataset than the BlobGAN trained on the IED-Coreset.

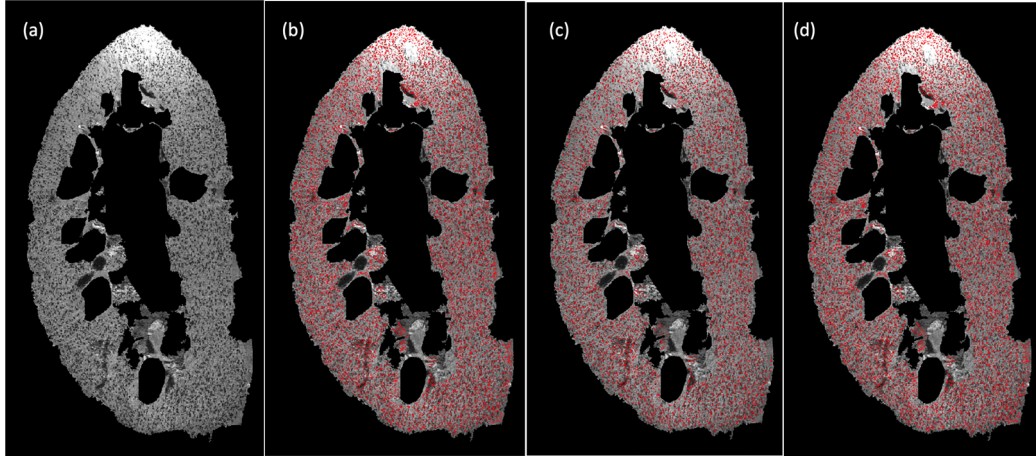

**Figure 3.** Glomerular segmentation results from 3D MR images of human kidney (CF1 slice 256). (**a**) Original magnitude image. (**b**) Glomerular segmentation results of BlobGAN trained on entire dataset (1000 random samples). (**c**) Glomerular segmentation results of BlobGAN trained on IED-Coreset ($k$ = 10). (**d**) Glomerular segmentation results of BlobGAN trained on FDD-Coreset ($k$ = 10).

### 4.4. Discussion: Clinical Translation

Imaging biomarkers from blob images have great potential to support clinical decisions. Taking glomerulus-based imaging biomarkers ($N_{glom}$, $aV_{glom}$) from Experiment II as an example, the biomarkers can serve as significant indicators in clinical trials, reducing cost and burden associated with related study on early detection of kidney diseases. While these biomarkers are clinically important, to date, the only methodology to obtain these biomarkers are destructive stereological approaches that can only be performed post mortem. Advanced development of magnitude resonance imaging (MRI) is making the measurement of these biomarkers more feasible under a non-destructive imaging approach. However, there are several challenges in glomerulus detection by MRI. First, the glomeruli are small relative to the imaging resolution and have a similar visual appearance as the noise. Second, there are over a million glomeruli in the cortex of the human kidney, and the image intensity distribution is heterogeneous. Third, a significant proportion of glomeruli overlap due to the low image resolution, making it difficult to identify them as individual blobs. The BlobGAN [8] has been developed to address these issues for improved glomeruli identification. However, the training of the BlobGAN is slow and computationally expensive. With the FDD-Coreset applied in the training of the BlobGAN, the training time of the BlobGAN is reduced significantly and the performance approximates the BlobGAN trained on the entire dataset. It shows the promise of rapid acquisition where imaging data can be used in a timeframe to influence patient care.

### 5. Conclusions

In this research, an FDD-Coreset approach is proposed to select subset samples used in GAN training to address the computational challenges. A Fréchet Descriptor Distance (FDD) is first proposed to measure the difference between each pair of blob images. Second, we select the Coreset from the entire dataset using FDD metric. We conducted two experiments. In the first experiment, we evaluated the performance of the FDD-Coreset on the BlobGAN using a set of 3D synthetic blob images. Computational time and eight performance metrics including Peak Signal-to-Noise Ratio (*PSNR*), Detection Error Rate (*DER*), Precision, Recall, F-score, Dice coefficient, Intersection over Union (*IoU*), and Blobness (*PB*), were used to compare the performance of the FDD-Coreset with the entire dataset (1000 random samples) and the IED-Coreset. Compared to training on the entire dataset, the BlobGAN trained on the FDD-Coreset greatly achieved a more than 96% decrease in computational time, a more than 25% increase in *PSNR*, a more than 40% decrease in *DER*, and provides comparable performance in blob detection (Precision, Recall, F-score), blob segmentation (Dice, *IoU*), and blob synthesis (*PB*). Three 3D MR human kidney images were studied in the second experiment. Compared to training on the entire dataset, the BlobGAN trained on the FDD-Coreset greatly achieved a more than 96% decrease in computational time and comparable performance in $N_{glom}$ and $aV_{glom}$. From the results of the two experiments, we conclude that the BlobGAN trained on the proposed FDD-Coreset significantly reduces the training time of the BlobGAN, achieves higher denoising performance (*PSNR*) and maintains approximate performance of blob identification compared to training on the entire dataset.

While the results of this study are encouraging, there is room for improvement. The proposed FDD is derived from object statistics, which are blob descriptors from blob distribution. This could be potentially used for accelerating other object-focused image translation models where the shape of objects (blob, nuclei, glomeruli, etc.) follow Gaussian distribution. However, if the shape of objects (tumor, organ, etc.) is irregular and the distribution is unknown, the proposed FDD metric will not be applicable. It is our plan to explore a distribution-agnostic-based Coreset approach as the next step. In addition, the blob images studied in this research have a large number of objects to derive the statistic descriptors. Some other medical image applications may have a smaller number of objects (e.g., tumors, lesions) to be investigated. We plan to study other discriminative features instead of shape as object statistics, to increase the effectiveness of Coreset.

**Author Contributions:** Conceptualization, Y.X. and T.W.; methodology, Y.X. and T.W.; validation, Y.X.; formal analysis, Y.X.; investigation, Y.X.; resources, T.W.; data curation, Y.X. and T.W.; writing—original draft preparation, Y.X. and T.W.; writing—review and editing, Y.X., T.W., J.R.C. and K.M.B.; visualization, Y.X.; supervision, T.W.; project administration, T.W.; funding acquisition, T.W., J.R.C. and K.M.B. All authors have read and agreed to the published version of the manuscript.

**Funding:** This research was supported by funds from the National Institute of Health award under grant number R01DK110622, R01DK111861.

**Institutional Review Board Statement:** In this research, three human kidneys were obtained after autopsy through a donor network (The International Institute for the Advancement of Medicine, Edison, NJ) after receiving Institutional Review Board (IRB) approval. Details of IRB are in [19].

**Informed Consent Statement:** Informed consent was obtained from Arizona State University and details are in [19].

**Data Availability Statement:** The data presented in this study are available on request from the corresponding author.

**Acknowledgments:** The U.S. Government is authorized to reproduce and distribute for governmental purposes notwithstanding any copyright annotation of the work by the author(s). The views and conclusions contained herein are those of the authors and should not be interpreted as necessarily representing the official policies or endorsements, either expressed or implied, of NIH or the U.S. Government.

**Conflicts of Interest:** The authors declare no conflict of interest.

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
