# Peer review of "GAN Training Acceleration Using Fréchet Descriptor-Based Coreset"

_applsci, doi:10.3390/app12157599_

Round 1
Reviewer 1 Report
In this paper, the Fréchet Descriptor Distance (FDD) coreset approach is offered to establish subset samples in GAN training to address the computational challenges. The FDD approach is proposed to measure the difference between each pair of blob images. The paper is well-written and organized, but there are still some concerns about the manuscript's quality improvement.
It looks reasonable enough to split your current Introduction into the new Introduction and the Related Works where you can put down the analysis of the current sources on the topic.
Some discussion must be added focusing on the explanation of the obtained outcomes.
It should additionally dwell on the fact of why the FDD distance is used in this case.
It looks like you do not use any similar GANs for comparison. You should add some outer comparisons with the analogs.
Is it also possible to share a link to the code of the developed GAN?
Reviewer 2 Report
I personally thank the authors for providing such a nice paper. The presented methodology is very interesting, and obtained result would be useful for the research community.
I have no special requirements since the paper is very well written and contains significant novelty. I will only appreciate it if maybe authors use the abbreviation for blobness in Formula 17.
Reviewer 3 Report
The authors propose Coreset method to accelerate the training process of GAN. The method is described clearly and useful according to the experimental results. There are a few comments as follows:
(1) The proposed method is mainly for blob identification, in the Introduction part, the related work on blob identification should be presented.
(2) The authors should give a clear description of GAN used in the paper both in methodology and experimental parts. How to train and fine tune the GAN to reach the best performance on blob data is also very important for researchers to evaluate the effectiveness of the proposed method, because only in this way, it is reasonable to compare and give conclusion the proposed method with other methods.
